# Identity abuse against sexual and gender minority communities: The Being LGBTQI+ in Ireland study

Carmel Downes[ID], Karin O'Sullivan, Jan de Vries[ID], Matt Kennedy[¤a], Renee Molloy[¤b], Aviejay Paul, Agnes Higgins[ID]*

School of Nursing and Midwifery, The University of Dublin Trinity College, Dublin, Ireland

¤aCurrent address: Winchester School of Art, University of Southampton, England
¤bCurrent address: School of Nursing and Midwifery, Monash University, Melbourne, Australia
* ahiggins@tcd.ie

## Abstract

Identity abuse targets people on the basis of immutable characteristics, and has the potential to cause greater emotional and psychological distress for this reason. People with multiple intersecting identities may be at greater risk. While the LGBTQI+ population experience higher rates of abuse than the general population, studies often focus on one type of abuse and rarely examine within LGBTQI+ group variation. A national cohort study involving over 2,800 LGBTQI+ participants aged 14+ examined the prevalence and predictors of LGBTQI+ identity abuse across six forms of abuse, as well as experiences of identity abuse more broadly via an open-ended question. Lifetime and past year prevalence of at least one form of abuse was 79% and 42% respectively. Polyvictimisation (two or more forms of abuse) was experienced by over half of the sample within their lifetime. Besides LGBTQI+ identity abuse, participants also experienced other forms of identity abuse, including ableism, racism, and sexism. LGBTQI+ people with disability were particularly at risk of abuse. Asexual and bisexual participants had lower odds of many forms of identity abuse. Risk, however, was constituted differently across the different forms of abuse, highlighting the importance of not siloing abuse in research by examining each type independently of one another. Interventions to tackle prejudice-based abuse are required, including those that address heteronormative and gender normative assumptions that seem to underpin prejudice, as well as tackling ageism and racism. Polices and measures to safeguard LGBTQI+ people against identity abuse also need to be enacted.

## Introduction

When abuse is directed at aspects of one's identity with the aim of devaluing those identities, exercising control over another person or perpetrating harm against them, this may be referred to as identity abuse [1,2]. Identity abuse is an umbrella term

**Data availability statement:** Data are not publicly available as ethical approval was granted to only share data within the research team and with third party parties where contractual arrangements are in place. However, requests for access to the underlying data may be submitted to the overseeing ethics body, subject to institutional review. To request access to the data, please contact the Faculty of Health Sciences Research Ethics Committee: ethics-committee@tcd.ie.

**Funding:** The study was funded by the Health Service Executive (HSE) (https://www.hse.ie/), the National Office for Suicide Prevention (NOSP) (https://www.hse.ie/eng/services/list/4/mental-health-services/nosp/), the HSE National Social Inclusion Office (NISO) (https://www.hse.ie/eng/services/list/5/publichealth/publichealthdepts/nsio/) and the What Works and Dormant Accounts Fund, Department of Children, Equality, Disability, Integration and Youth (https://whatworks.gov.ie/) through Belong To Youth Services (https://www.belong-to.org/). The recipient of the funding was the Principal Investigator, Professor Agnes Higgins. The funders had no role in study design, data collection and analysis, decision to publish, or preparation of the manuscript.

**Competing interests:** The authors have declared that no competing interests exist.

used to describe a range of harmful behaviours directed towards a person because of their identity. It can take the form of verbal threats, physical assault, and sexual violence [3]. Rooted in heteronormative, cisnormative, racial, ageist and ableist assumptions, identity abuse may target aspects such as gender identity, sexual identity, race, age and physical or mental ability [4]. What distinguishes identity abuse from other forms of abuse is that it stems specifically from bias, prejudice and hatred of a person's immutable identity [5]. Examples include, hate speech – offensive material intended to degrade a particular social group [6], and microaggressions, which undermine members of marginalised groups through biased actions, whether conscious or subconscious [7].

Identity abuse against LGBTQI+ people on the basis of their gender and sexual identity is on the rise throughout Europe and internationally, with hate speech an increasing feature of public media and online discourse in relation to transgender populations [8,9]. This ever-present risk prompts some LGBTQI+ people to truncate or hide their identities or modify their behaviours in order to safeguard their safety [10,11].

The concept of identity abuse as it relates to LGBTQI+ identity has been studied in the context of intimate partner violence (IPV) [4,12]. In these studies, conducted in the US via an online survey, practices which are categorised as LGBTQI+-related identity abuse include employing homophobic or transphobic language, undermining and belittling one's LGBTQI+ identity, isolating a partner from their connections in the LGBTQI+ community, and threats to 'out' the person's gender, sexuality or HIV status. Findings from these studies highlight an association between identity abuse and symptoms of depression and Post Traumatic Stress Disorder, with transgender and gender non-conforming participants experiencing higher rates than cisgender participants, while queer or bisexual participants were more likely to experience it compared to lesbian and gay participants.

LGBTQI+ identity related abuse has also been researched, although less extensively, in the context of family violence. An Australian study, which surveyed young people (aged 16–20) through a mixture of both closed and open-ended questions, found that LGBTQI+ identity abuse co-occurred with other forms of family violence including verbal abuse, physical abuse and sexual abuse [13]. This study considered policing of gender norms and forced conversion practices as indicators of identity abuse. The young participants identified many adverse immediate and long-term effects, including emotional and mental health difficulties, physical injuries not only from the abuse but also from self-harm and eating disorders, school difficulties related to poor attendance and performance, issues around forming social connections, and missing out on cultural experiences related to not being able to be their true selves.

LGBTQI+ identity abuse has also been studied in other settings, although not conceptualised as identity abuse. In online settings, experiences of gender-and sexual identity based harassment via electronic communications devices were examined in a survey of LGBT adults, a subset of a larger sample of Australian and British adults [14]. It was defined as "offensive comments, threats or other harassment directed at

an individual's gender or sexual identity" (p.200), and transgender individuals were found to experience higher rates than heterosexual cisgender individuals. A workplace survey of gender-based harassment experienced by LGBQ employees in non-coastal regions of the U.S found that gender harassment was a relatively common experience and tended to go hand-in-hand with heterosexist harassment [15]. In this study, LGBQ people exposed to high levels of gender harassment in the form of sexism and policing of gender non-conformity experienced job burnout and dissatisfaction.

In the school context, gender and sexual identity-based harassment was examined through secondary analysis of the California Healthy Kids Survey [16]. It was found to be higher for transgender adolescents compared to cisgender adolescents, and among adolescents who were lesbian, gay, or bisexual compared to heterosexual. Both forms of harassment increased the likelihood of cigarette use (i.e., vaping), cigarette smoking, alcohol use and heavy episodic drinking. Gender abuse has also been linked to an increased risk of HIV and symptoms of depression in a 3-year prospective study of risk factors among male-to-female transgender individuals in New York [17].

Together these aforementioned studies point specifically to the vulnerability of transgender populations to gender-based abuse [4,14,17], due to nonconformity and perceived breaches of gender norms [18]. Research with college students in the US also found that transgender populations are at increased risk of multiple forms of victimization, including IPV, sexual violence and physical assault compared to cisgender populations [19], which has been shown to contribute to more pronounced mental health issues than experiencing one type of victimisation alone [17,20,21].

Members of the sexual and gender minority community belong to multiple social groups and have multiple intersecting identities that may shape their exposure to different and overlapping types of abuse. Using the lens of intersectionality theory, researchers have highlighted how LGBTQI+ people with a disability are more at risk of harassment than those without a disability [22,23]. Likewise, LGBQ adults from a racial and ethnic minority background experience higher rates of harassment than those from a white background [24]. Among the transgender and gender non-conforming population, an increased risk of sexual victimization and IPV victimization has been found among those with a disability [19] while another study found that transgender people from ethnic minority backgrounds with a disability experienced greater discrimination than those from ethnic majority backgrounds without a disability [25]. US based research suggests that LGBTQ+ young people living rurally are more at risk of abuse and discrimination, with the conservative nature of rural areas being cited as well as a lack of support and resources [26,27]. While LGBTQ+ people in Ireland identify a lack of supports and services in rural communities [28], it is not clear whether rural versus urban living impacts one's experience of LGBTQI+ identity abuse. From a minority stress perspective, identity abuse can exacerbate feelings of isolation, internalised shame/stigma and psychological distress [29]. Some research also hints at differences between sexual identities in their experience of identity abuse, although greater exploration is needed taking account of diverse sexual identities that often receive less attention in research, including pansexual and asexual individuals [30,31]. In an effort to better understand issues of identity abuse, in this paper we explore within-group variation in exposure to a broad range of identity abuses taking account of age, gender, sexual orientation, disability, urbanicity and ethnicity.

## Methods

The paper is based on data from the *Being LGBTQI+ in Ireland* study, which was a national cohort study of the wellbeing and mental health of LGBTQI+ participants in Ireland [28]. The study was conducted via an anonymous online survey hosted on Qualtrics (Qualtrics©, 2020). Inclusion criteria were any person who identified as LGBTQI+, was 14 years of age or over and living in the Republic of Ireland. The survey was designed by the research team in conjunction with the Research Advisory Group (RAG) involving people from the sexual and gender minority community. This consultative process helped to ensure the relevance and appropriateness of the research questions, thus enhancing the validity of the study [32,33]. Although the primary focus was on wellbeing and mental health, a range of other topics were explored, including identity abuse, which is the primary focus of this paper.

## Aims

This paper has the following aims:

- To examine the prevalence of types of LGBTQI+ identity-related abuse among LGBTQI+ participants including verbal abuse, online abuse, physical abuse, sexual abuse, threat of outing and gender abuse in their lifetime.

- To examine how ethnicity, sexual orientation, gender identity, age, disability status and urbanicity influence an individual's likelihood of having experienced various forms of LGBTQI+ identity-related abuse, including verbal abuse, online abuse, physical abuse, sexual abuse and threat of outing in their lifetime.

- To analyse free text responses to gain possible insight into other forms of identity abuse experienced by participants.

## Recruitment of sample

A multipronged approach was taken towards recruitment, which included publicising the study through several mediums in order to reach as many diverse LGBTQI+ participants as possible. In addition to displaying posters in spaces frequented by hard-to-reach populations, adverts were placed in regional print and digital media (newspapers and news and media sites), a member of the RAG was interviewed on regional radio stations to promote the study, and the study was publicised through both paid and unpaid content on social media platforms (incl. Facebook, Instagram, twitter, snapchat, tiktok) using social media assets (posters, graphics, banners etc.) created for the study. Study information was disseminated to networks and LGBTQI+ organisations as well as more general organisations in the community, education and health sectors. These networks were requested to similarly promote the study online using social media assets furnished to them.

## Data collection

Experiences of LGBTQI+ identity-related abuse were assessed using seven items adapted from a previous survey of LGBT participants in Ireland [34]. All participants were asked about verbal, physical, sexual, and online abuse, as well as threat of outing, with transgender and gender non-conforming participants asked about misuse of name/pronouns. These were conceptualised as forms of identity abuse as participants were asked whether they experienced them due to their LGBTQI+ identity. The available response options were: Within the last month; Between a month and six months ago; Between six months and a year ago; More than one year ago; Never. For the purpose of analysis, the categories were merged to determine past year prevalence and lifetime prevalence.

   Additional data collected included demographics and background information, such as, age, gender identity, sexual orientation, ethnicity, area living (rural vs. urban) and disability status. This latter measure was sourced from Ireland's Census 2022 [35] and relied on self-reported identification of having a disability, defined as having any long-lasting conditions or difficulties in relation to visual impairment, hearing impairment, physical disability, intellectual disability, learning disability, psychological conditions and chronic conditions. For the purpose of analysis, disabilities were categorised into four groups: no disability; physical health related disability, which included visual impairment, hearing impairment, physical disability, and chronic condition; mental health/neurodevelopmental related disability, which included intellectual disability, learning disability and psychological disability; and both physical and mental health/neurodevelopmental related disability. Gender identity was measured by asking participants: 1) how they identified their gender, with four response options available: i) man; ii) woman; iii) non-binary; iv) not listed: please describe, and 2) whether their gender identity matched the sex they were assigned at birth. Participants who responded 'no' to this latter question were then asked if they identified as transgender. Based on the responses to these gender questions, a new gender variable was created consisting of three categories: cisgender women, cisgender men, and trans and gender non-conforming. Participants were also asked to indicate if they had ever experienced violence and harassment for reasons other than because of their LGBTQI+ identity,

for example because of race or ethnicity or disability, and those that had were invited to comment further in a free text box provided, if they felt comfortable to do so. No word limit was attached to the free text response. Participants self-selected into the survey and sample representativeness is unknown given that there is no reliable estimate of the LGBTQI+ population in Ireland. Survey data were collected over a seven-week period between 13 September 2022 and 31 October 2022.

### Ethical approval

Ethical approval was granted from the Research Ethics Committee of the Faculty of Health Sciences in Trinity College Dublin (ID No. 220505). Informed written consent was obtained from participants. The anonymous online survey included a participant information leaflet (PIL) and a consent form which participants had to complete online prior to participation. The PIL also directed participants towards the study website which hosted a list of support services should they require support. The requirement for parental consent for participants aged under 18 was waived in order not to risk excluding those who may have wanted to participate but may not have wanted to disclose their LGBTQI+ identity to their parents/ guardians, which could potentially bias the results. All data files were password protected and stored in accordance with Data Protection legislation.

### Data analysis

Quantitative data were analysed using SPSS Statistics Version 27 [36]. Frequencies and percentages were generated to describe the data. The valid percentage (after missing data excluded) is reported. Bivariate analysis between the predictor variables (age, gender identity, sexual orientation, ethnicity, disability status, and area living) and the outcome variables (six items relating to lifetime experience of LGBTQI+ identity-related abuse) was carried out using the Chi-square test. Predictor variables that were found to have significant associations (i.e., p-value < .05) in the bivariate analysis were then entered into a multivariate logistic regression model to control for other predictors. Separate logistic regression models were run for each of the abuse items. A p-value of less than 0.05 was used to identify significant associations.

A qualitative descriptive approach was used to analyse free text comments. Qualitative descriptive research is a methodological approach that seeks to provide a straightforward account or description of the participants' experiences [37]. Grounded in naturalistic inquiry, its focus is on low-inference interpretation rather that extensive theoretical abstraction [38] and is particularly useful in generating insights to inform practice and policy [39]. To commence the analytical process, data were cleaned to remove any identifying information such as names of people/organisations/places, and imported into QSR NVivo Version R1 [40]. Two members of the research team (KO'S and AH) read the qualitative data and developed a coding framework based on the types of identity abuse experienced (i.e., gender, ethnicity, disability, age), with the option to add a new code to address any emerging ideas. Following this, using the framework agreed, the data were deductively coded by one researcher (KO'S). Partway through the analysis, in discussion with a second researcher (AH), additional codes called 'invisible disability' and 'intersectionality' were added to accommodate emerging ideas. The final analyses was also reviewed by the second researcher (AH) and the exemplars used within the paper agreed by both researchers, enhancing the validity and reliability of the process.

## Results

In total, 2806 participants provided responses to the survey. Nearly a third (30.2%) identified as transgender and gender non-conforming (TGNC), with the remainder being cisgender men (32%) and women (37.8%). A third (33.9%) identified their sexual orientation as gay, nearly a quarter (23.3%) as bisexual, approximately one fifth (19.7%) as lesbian while the remainder comprised of those who identified as queer (12.5%), asexual (4.1%) and pansexual (6.5%). The majority of the sample (69%) was aged 35 and under, was predominantly urban with 70% living in either a town, suburb or city, and overwhelmingly White Irish (83%). 69% reported some form of disability. Table 1 shows the socio-demographic profile of the sample.

**Table 1. Sample profile.**

| Characteristic | Category | n | % |
|---|---|---|---|
| Gender identity | Cis men | 892 | 32.0 |
| | Cis women | 1053 | 37.8 |
| | TGNC | 841 | 30.2 |
| Sexual orientation | Gay | 887 | 33.9 |
| | Lesbian | 514 | 19.7 |
| | Bisexual | 608 | 23.3 |
| | Queer | 327 | 12.5 |
| | Asexual | 108 | 4.1 |
| | Pansexual | 169 | 6.5 |
| Age | 14-18 | 631 | 22.7 |
| | 19-25 | 560 | 20.2 |
| | 26-35 | 729 | 26.3 |
| | 36-45 | 424 | 15.3 |
| | 46+ | 433 | 15.6 |
| Ethnicity | White Irish | 2332 | 83.1 |
| | White Non-Irish | 322 | 11.5 |
| | Black, Asian, Mixed or Other | 151 | 5.4 |
| Area living | Rural | 721 | 27.6 |
| | Urban | 1890 | 72.4 |
| Disability status | No disability | 799 | 30.6 |
| | Mental health/neurodevelopmental related disability | 691 | 26.5 |
| | Physical health related disability | 273 | 10.5 |
| | Both physical and mental health/ neurodevelopmental related disability | 846 | 32.4 |

## LGBTQI+ identity-related abuse

Nearly four fifths (78.8%, n = 2047) of the sample experienced at least one form of abuse due to their LGBTQI+ identity (verbal abuse; online abuse; physical abuse; sexual abuse and threat of outing) in their lifetime. 53.8% had experienced two or more forms of abuse related to their LGBTQI+ identity, i.e., polyvictimisation (Fig 1).

Table 2 displays the lifetime and past year prevalence rates of LGBTQI+ identity-related abuse across six forms of abuse. Verbal abuse was the most prevalent form of abuse encountered with a 72% lifetime rate. The next most prevalent form of abuse for the whole sample was being threatened with being outed, with one third experiencing this in their lifetime. This was closely followed by being touched sexually without consent (30.2%) and social media harassment (28.3%). Both verbal abuse and social media abuse were the most prevalent past-year abuse types among the whole sample. The majority of TGNC participants had experienced gender abuse (being misnamed/misgendered), with nearly three in five experiencing it in the last year alone. 41.9% experienced at least one form of abuse due to their LGBTQI+ identity (verbal abuse; online abuse; physical abuse; sexual abuse and threat of outing) in the previous year (n = 1,088) while 19% experienced two or more forms of abuse related to their LGBTQI+ identity (i.e., polyvictimisation) in the previous year.

## Predictors of identity abuse

Table 3 shows the results of the multivariate analysis of the association between lifetime prevalence of six types of abuse and socio-demographic characteristics. As the bivariate analysis showed no statistically significant association between

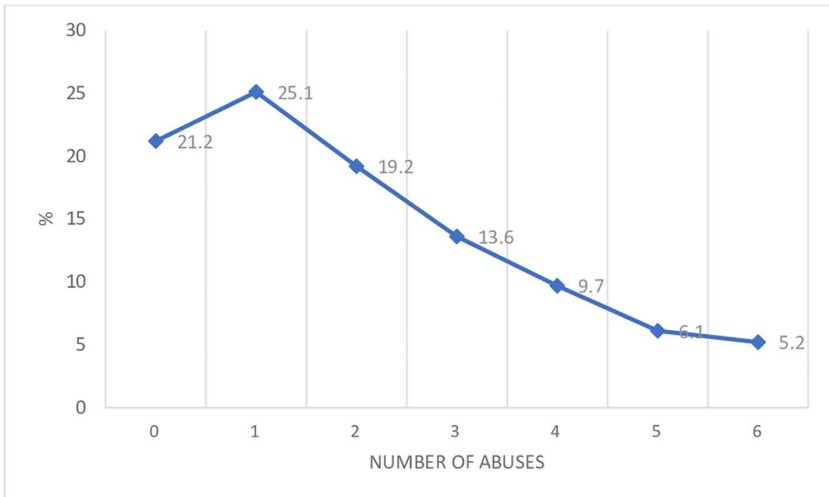

**Fig 1. % Prevalence of number of abuses experienced.**

**Table 2. Prevalence of lifetime and past year identity-related abuse.**

| Types of abuse | Item | % Lifetime | % Past Year |
|---|---|---|---|
| Verbal abuse | Verbally hurt you because you are LGBTQI+ (e.g., made fun of you, insulted you) | 71.8%, n = 1923 | 35.7%, n = 955 |
| Online abuse | Wrote hurtful things about you on social media because you are LGBTQI+ (e.g., Twitter, Facebook or Instagram) | 28.3%, n = 752 | 13%, n = 344 |
| Physical abuse | Punched, hit, or physically attacked you because you are LGBTQI+ | 24.1%, n = 646 | 5.2%, n = 139 |
| Sexual abuse | Attacked sexually because you are LGBTQI+ (e.g., raped, made to have an unwanted sexual experience) | 16.5%, n = 438 | 3.5%, n = 94 |
| | Touched in a sexual manner without your consent because you are LGBTQI+ | 30.2%, n = 810 | 8.9%, n = 239 |
| Threat of outing | Been threatened by another person to out you as LGBTQI+ | 33.1%, n = 885 | 8.9%, n = 238 |
| Gender abuse | Purposely used the wrong name/pronoun when talking about you* | 68%, n = 517 | 59.7%, n = 454 |

Note: *Gender abuse item was only displayed for participants identifying either as non-binary or trans.

ethnicity and identity abuse, it was not entered into the multivariate logistic regression model. Across all abuse types, the odds were higher among those with a mental health/neurodevelopmental related disability and among those with both a physical and mental health/neurodevelopmental related disability compared to those with no disability. The additional risk factors varied across abuse types. TGNC participants had higher odds of experiencing verbal abuse compared to cis men while bisexual, asexual and pansexual participants had lower odds compared to lesbian participants. Similarly, TGNC participants had higher odds of experiencing online abuse compared to cis men while cis women had lower odds than cis men. Participants aged 26–35 had higher odds of online abuse compared to participants aged 46 + . The factors associated with higher odds of experiencing physical abuse were identifying as TGNC, having a mental health/neurodevelopmental related disability, and identifying as gay. Lower odds of experiencing physical abuse were associated with being aged under 35 compared to 46 + , identifying as a cis woman compared to a cis man, and identifying as asexual compared to identifying as lesbian. Lower odds of being sexually attacked was associated with identifying as a cis woman compared to cis man and being aged under 25 compared to being 46 + years. Those aged 14–18 years and cis women also had lower

odds of being touched in a sexual manner without consent along with those identifying as bisexual and asexual. Higher odds of being touched in a sexual manner without consent was associated with being aged 26–45 compared to 46+years. Lower odds of being threatened with outing was associated with identifying as bisexual, queer, and asexual compared to lesbian.

## Experience of multiple forms of identity abuse

In total 690 (33.9%) participants reported experiencing identity abuse for reasons other than because of their LGBTQI+ identity, with 587 providing responses to the free text question on other identity-related reasons for abuse, including gender, race/nationality and disability.

Gender related abuse was primarily cited by cisgender women and was considered a common experience. Cisgender women described being subjected to misogynistic comments, being physically touched without consent, as well as sexual harassment and assault.

*"As a woman I have experienced harassment many times. Street harassment, unwanted touch, aggression if I don't response positively to unwanted attention, even threats. It gets less as I get older, but it's still there" (44, woman, lesbian, ID2778).*

*"Harassed quite regularly in day-to-day situations as a woman. Belittling, direct insults, unwanted physical contact, cat calling* [a form of street harassment that involves unwanted sexualised comments or gestures] *etc." (29, woman, pansexual, ID2538)*

*"I am a woman. I have experienced more sexual harassment, micro aggressions and unwanted touching/groping as well as more serious issues than I could ever list in a box." (28, woman, bisexual, ID2286)*

Abuse rooted in racism and xenophobia was also reported. Mostly participants reported verbal attacks but instances of physical assault were also reported and attributed to differences in one's skin colour, one's verbal accent, and speaking in one's native language.

*"I got physically attacked twice (separate occasions) because of my race." (17, non-binary, lesbian, ID2000)*

*"I have been called racist slurs, had racist comments shouted at me, and had racist gestures done to me in public." (15, transman, asexual, ID1603)*

*"Even though I am white Irish by heritage, I experienced more racism in Ireland than homophobia. I tan [skin goes browns] easily and therefore people assume I am not Irish. I have been called paki [ethnic slur for a person from Pakistan or South Asia], raghead [terms of abuse for person who wears a turban keffiyeh], asked if I was an asylum seeker etc." (36, man, gay, ID280)*

*"I was bullied a lot in primary school, I was the only person in my class with an ethnic background and everyone else was white. I would be called names, made fun of for what I ate, what I looked like, the language my family spoke and how it sounded, stereotypes..." (15, woman, ID1295).*

Disability and ableism were also cited as reasons for abuse. Particularly prominent disabilities referenced in relation to abuse were neurodivergent disabilities, such as autism. In addition to experiencing abusive behaviours, many described experiences of '*belittling*', '*infantilising*', being made to feel '*stupid*' or '*dumb*', being '*mocked*', and being '*shamed*'.

*"I am autistic and received sexual harassment because of it, with the perpetrators claiming I was "too dumb to realise it." (18, transman, asexual, ID2387)*

**Table 3. Multivariate analysis of association between forms of abuse and socio-demographic characteristics.**

| Participant characteristic | Categories | Verbal abuse | Online abuse | Physical abuse | Sexual abuse | | Threat of outing |
|---|---|---|---|---|---|---|---|
| | | Verbally hurt you because you are LGBTQI+ | Wrote hurtful things about you on social media because you are LGBTQI+ | Punched, hit, or physically attacked you because you are LGBTQI+ | Attacked sexually because you are LGBTQI+ | Touched in a sexual manner without your consent because you are LGBTQI+ | Been threatened by another person to 'out' you as LGBTQI+ |
| | | Odds, 95% CI | Odds, 95% CI | Odds, 95% CI | Odds, 95% CI | Odds, 95% CI | Odds, 95% CI |
| Age | 14-18 | / | 1.15 (0.81-1.64) | 0.30*** (0.21-0.44) | 0.45*** (0.29-0.69) | 0.48*** (0.33-0.68) | 1.29 (0.94-1.78) |
| | 19-25 | / | 1.33 (0.93-1.89) | 0.41*** (0.28-0.58) | 0.66* (0.44-1.00) | 0.99 (0.71-1.37) | 0.98 (0.72-1.34) |
| | 26-35 | / | 1.91*** (1.39-2.63) | 0.73* (0.54-0.98) | 1.17 (0.81-1.69) | 1.48** (1.10-1.98) | 0.93 (0.70-1.24) |
| | 36-45 | / | 1.05 (0.73-1.52) | 0.87 (0.63-1.20) | 1.21 (0.81-1.81) | 1.31 (0.95-1.80) | 0.86 (0.62-1.18) |
| | 46+ | Ref. | Ref. | Ref. | Ref. | Ref. | Ref. |
| Gender identity | Cis man | Ref. | Ref. | Ref. | Ref. | Ref. | Ref. |
| | Cis woman | 0.80 (0.60-1.08) | 0.56*** (0.44-0.72) | 0.42*** (0.30-0.59) | 0.56*** (0.42-0.74) | 0.59*** (0.43-0.79) | 0.87 (0.65-1.17) |
| | TGNC | 1.73*** (1.24-2.42) | 1.70*** (1.32-2.20) | 1.42* (1.01-1.99) | 0.99 (0.73-1.35) | 1.04 (0.76-1.42) | 1.34 (0.99-1.82) |
| Sexual orientation | Lesbian | Ref. | Ref. | Ref. | Ref. | Ref. | Ref. |
| | Gay | 1.14 (0.80-1.63) | / | 1.57* (1.06-2.32) | / | 1.40 (0.99-1.98) | 1.23 (0.88-1.71) |
| | Bisexual | 0.50*** (0.37-0.67) | / | 0.76 (0.52-1.10) | / | 0.63** (0.46-0.85) | 0.71* (0.54-0.94) |
| | Queer | 1.05 (0.72-1.53) | / | 1.08 (0.72-1.60) | / | 0.84 (0.59-1.20) | 0.70* (0.50-0.98) |
| | Asexual | 0.28*** (0.17-0.46) | / | 0.49* (0.25-0.96) | / | 0.49* (0.27-0.87) | 0.35*** (0.21-0.61) |
| | Pansexual | 0.61* (0.39-0.94) | / | 1.29 (0.80-2.08) | / | 1.12 (0.73-1.73) | 0.89 (0.59-1.34) |
| Area living | Rural | / | / | Ref. | / | Ref. | / |
| | Urban | / | / | 1.26 (1.00-1.60) | / | 1.08 (0.87-1.33) | / |
| Disability status | No disability | Ref. | Ref. | Ref. | Ref. | Ref. | Ref. |
| | Mental health/neurodevelopmental related disability | 1.57*** (1.23-2.01) | 1.51** (1.15-1.98) | 1.55** (1.16-2.07) | 2.56*** (1.81-3.61) | 1.99*** (1.53-2.58)*** | 1.37* (1.06-1.76) |
| | Physical health related disability | 1.37 (0.99-1.88) | 1.23 (0.85-1.76) | 1.36 (0.95-1.94) | 1.35 (0.84-2.16) | 1.28 (0.92-1.79) | 1.12 (0.81-1.55) |
| | Both | 2.26*** (1.75-2.92) | 2.23*** (1.71-2.90) | 2.35*** (1.78-3.11) | 4.18*** (3.00-5.83) | 2.70*** (2.08-3.50) | 2.11*** (1.66-2.70) |

i. '/' represents not applicable since that variable did not have a significant association in the bivariate analysis and was therefore not entered into the multivariate analysis.

ii. 'Ref.' indicates the category within the variable to which the other categories are compared.

iii. Odds with 95% confidence intervals in brackets.

iv. Levels of significance: * p < 0.05, ** p < 0.01, *** p < 0.001.

v. Green shaded boxes signify a statistically significant result.

*"I am treated differently because I'm a wheelchair user. I often experience abuse when I'm taking public transport to work or in healthcare settings." (48, trans non-binary, pansexual, ID88).*

Invisible disabilities were also mentioned in relation to receiving abuse. In this context, participants felt verbally attacked when people failed to recognise and accommodate their disability, and challenged its authenticity.

*"I have an invisible disability… so people don't believe and refuse to accommodate me. Both medical professional and teachers, etc at school don't believe I'm ill and think I do it for attention." (18, woman, queer, ID1753)*

*"As someone with an "invisible" disability I have had MANY people demand to know how I have the right to a travel pass. This usually includes "but you look normal" or "you don't look like you have anything wrong with you" Some people have been very verbally aggressive with this..." (35, woman, pansexual, ID420).*

While most of the comments didn't allude to the intersectionality of identities, some participants commented on how being a woman and queer, or being autistic and queer compounded the abuse they experienced while others named several aspects of their identity that were targeted.

*"Because of my autism, I've experienced social exclusion and bullying my entire life, which wasn't helped by my queerness." (20, genderqueer, queer, ID1836)*

*"Bullying and discrimination and regular microaggressions relating to my ethnicity/race and also my autism." (19, non-binary, asexual, ID1234)*

*"I am fat, disabled and femme presenting. I have had all manner of slurs shouted at me over the years." (30, non-binary, pansexual, ID2395)*

*"Just harassment and unwanted sexual attention, due to sexism and ableism." (32, questioning, queer, ID2372)*

## Discussion

When someone is abused on the basis of an innate part of their identity that they cannot change, it can cause greater adverse psychological effects for victims in comparison to victims of harassment and abuse which is not rooted in prejudice [41]. Thus, identity-based abuse has important consequences for wellbeing and mental health. This paper finds that the prevalence of LGBTQI+ identity related abuse is high, with nearly four-fifths of participants experiencing at least one form of abuse in their lifetime. While the lifetime prevalence (79%) reported in this study is high, the past-year prevalence is lower (42%) than what is reported within the literature, with an EU study finding that 54% of EU respondents experienced some form of harassment in the past year because they are LGBTQI+, with a similar figure of 52% reported for Ireland [9]. Polyvictimisation was also high among participants, with just over half of participants experiencing multiple forms of abuse due to their LGBTQI+ identity in their lifetime and one fifth experiencing it within the last year. With polyvictimisation associated with poorer physical and mental health outcomes among sexual and gender minorities [21,42], this is especially concerning as polyvictimisation more so than monovictimisation not only highlights the extent of people's experiences of harassment and stigma but ultimately contributes to more adverse mental health outcomes [43].

The predictive analysis further highlighted how disability and specific forms/combinations of disability were significant contributors to all forms of identity abuse against the LGBTQI+ sample, with a greater likelihood of experiencing abuse among those with a mental/neurodevelopmental health related disability compared to those without a mental/neurodevelopmental health disability, and an additional increased likelihood of experiencing abuse among those with both a mental/

neurodevelopmental health and physical health related disability compared to those without disability. That both physical and mental/neurodevelopmental health related disabilities significantly increases one's likelihood of experiencing abuse adds to the current evidence base, as to date there has been little exploration of experiences of LGBTQI+ identity related abuse among LGBTQI+ people with disability [22]. In the qualitative comments, while few participants explicitly identified intersectional experiences of harassment, many did recount abuse associated with their neurological variance. Research with adults on the autistic spectrum found intersections between disability, gender and sexuality in participants' experiences of harassment, and argued that it was not their identities per se that contributed to the harassment but the non-conforming ways in which they embodied their identities, such as not aligning with feminine or masculine expectations of their gender and exhibiting 'deviant' behaviours due to their neurological variance [44]. Our findings also revealed how invisible disabilities were often misunderstood and delegitimated, and subject to ridicule and abuse. LGBTQI+ people with invisible disabilities may not only have to consider issues of safety in relation to disclosing their sexual and gender identity, but may also face additional concerns around disclosing their disability. They may anticipate that it may be negatively perceived or rejected, as invisible disabilities are less socially included than visible disabilities [45].

With regard to gender, TGNC participants had an increased likelihood of verbal abuse, online abuse and physical abuse compared to cis men. Several studies show that transgender people experience more harassment and discrimination than cisgender people [46]. An EU survey of LGBTQI participants found that transgender and intersex respondents reported the highest levels of experiencing a physical and/or sexual attack for being LGBTQI in the past five years [9]. Similarly, a US survey of transgender participants found high levels of harassment due to gender identity or gender expression, with 30% reporting verbal harassment and 39% reporting online harassment in the last year [47]. Abuse against TGNC people is rooted in cisgenderism and gender normativity, which regard deviations from binary gender as abnormal [48]. Intentional misnaming and misgendering were common experiences for transgender and non-binary participants, with nearly three fifths reporting that they experienced it in the last year. The high rates of misgendering or identity invalidation is a concern as it is a unique form of minority stress for transgender and non-binary people which affects their emotional, social and mental wellbeing [49]. Not only does misgendering erase and makes invisible people's identities, but it challenges people's sense of agency over their gender identity, which has negative emotional and mental health impacts [50]. Misgendering also has the potential to invoke shame and distress, as well as increasing worry and fear about receiving unwanted negative attention if they correct people [48,49]. Intentionally misgendering people is also an example of the way in which people try to reinforce cisgenderism as well as communicate opposition to trans or non-binary people's identity [50].

Research on the relationship between age and harassment suggests that the risk diminishes with increasing age, but that it is also context specific, with workplace harassment more likely to increase with age [24,46]. Our study also reveals a complex relationship between age and type of abuse. Participants under 25 years had lower odds of experiencing physical or sexual abuse compared to those aged 46+, perhaps because older people over the course of their lifetime have encountered these forms of abuse. Indeed, high rates of lifetime physical assault have been found among older LGB adults [24]. On the other hand, those in the mid-20s to mid-30s had higher odds of experiencing LGBTQI+ identity abuse online and being touched sexually without consent, perhaps because of being higher users of social media [51]. They are also the age group who may be more active on the social scene (bar/clubs) which potentially increases the risk of sexual harassment.

While it is well known that LGBTQI+ people experience higher rates of violence and harassment compared to the general population [52,53], the risk of identity abuse against diverse and emerging sexual identities is under explored. With regard to sexual orientation, gay participants, most of whom were men, had higher odds of physical abuse. Research suggests that femmephobia (i.e., negative attitudes toward femininity in men) strongly predicts anti-gay behaviours (avoidant behaviour, verbal abuse, physical assault) among heterosexual men [54]. Others hypothesise that gay men may exhibit more gender atypical appearance or behaviour which may be deemed impermissible due to gender roles for men being

viewed in more stringent and less flexible terms than women's gender roles [55]. Research also suggests that atypicality in terms of individuals' appearance and behaviours is a risk factor for victimisation and this affects not only gender minorities but also sexual minorities [44,56,57]. While bisexual people have been found to be more vulnerable to identity abuse within their intimate relationships [4,58], and bisexual stigma and stereotypes found to contribute to a higher risk of abuse [59,60], our study found a reduced odds of verbal abuse, threats of being outed and being touched in a sexual manner without consent for bisexual participants.

In terms of asexuality, there is some research suggesting that people identifying as asexual may be less exposed to interpersonal harms than other sexual identities [61], with lower rates of verbal abuse, harassment and victimisation reported compared to LGB people [62]. Our findings are in line with these studies as lower odds of verbal abuse, threats of being outed and being touched in a sexual manner without consent were found for asexual participants. Lower rates among people who identify as asexual may be due to the relative invisibility of asexual identities and their ability to pass as heterosexual [60,61].

Interestingly, ethnicity was not significant in the bivariate analysis so was not entered into the multivariate analysis. This finding was surprising, given the research that indicates that LGBTQI+ people of colour experience higher rates of victimisation in various contexts in comparison to their white counterparts [24,62]. An EU survey found that LGBTIQ people with multiple intersecting identities, including ethnic or migrant backgrounds, experienced higher levels of discrimination than those without these backgrounds [9]. The lack of significance may be due to not having an adequate sample to accurately assess differences in abuse exposure across ethnicity. However, the qualitative data provided various accounts of identity abuse related to racism/xenophobia, such as difference in colour, verbal accent, dress, and use of language other than English.

## Strengths and limitations

The strengths of this study lie in being able to examine LGBTQI+ identity-related abuse among the LGBTQI+ population using a large sample size. Not only did the study examine multiple forms of LGBTQI+ identity related abuse, it also explored abuse related to other social identities. The involvement of members of the LGBTQI+ community on the RAG follows best practice in researching minority communities and enhances the validity of the research study. However, there are also a number of limitations to bear in mind. While the survey captured different forms of abuse, it may have omitted more covert or subtle forms of LGBTQI+ identity abuse, such isolating someone from their family or community due to their sexual or gender identity, forcing someone to conform to gender norms or undergo conversion therapy, or invalidating someone's identity by saying it is a phase or isn't real. Furthermore, there was little context around the abuse in terms of where it took place (with the exception of the question on online abuse) and who were the perpetrators, information which may have provided more understanding on the risks for LGBTQI+ populations and the kinds of interventions required. Additionally, the representativeness of the sample and the generalisability of the findings cannot be determined given that there is no reliable estimate of the LGBTQI+ population in Ireland. The self-selecting nature of participation, the recruitment strategies used, and the online hosting of the survey may have biased the sample and excluded those without an adequate level of digital literacy, those without access to a smart phone, device and/or internet, those who were not out, and those not as interested in LGBTQI+ issues or connected to LGBTQI+ communities.

## Implications for research

Results of this study point to several important avenues for future research. Interventions exist which aim to reduce gender identity prejudice including gender neutral policies and contact interventions which promote interaction between those who hold prejudices towards certain groups or individuals and those who belong to stigmatised minority groups. More research however is needed to understand the ways in which gender normativity and heteronormativity may underpin identity abuse and into interventions that can counter abuse, including evaluating the efficacy of such interventions [56].

Furthermore, when minority sexual and gender identities intersect with racial, ethnic, disability and other background characteristics, the risk of exposure to identity abuse may be heightened. Although ethnicity was not associated with any of abuse experiences, perhaps owing to the relatively small number of ethnic minority participants, disability emerged as a strong risk factor for abuse among the LGBTQI+ participants. As the researchers coded types of disability into two broad categories of physical and mental health/neurodevelopmental disability, a more nuanced approach examining specific disabilities, such as neurodivergent disabilities which were frequently cited as a target of harassment, should inform future research in the area of LGBTQI+ identity abuse against LGBTQI+ people with disability. Furthermore, future research should adopt an intersectional perspective when exploring identity abuse. Intersectionality theory [63] posits that multiple interlocking social identities produces and maintains a system of oppression or privilege, and it is therefore a critical lens for understanding the experiences and risk factors for people experiencing identity abuse. Similarly, minority stress theory recognises the cumulative effects of harassment and victimization among people who hold multiple stigmatised identities [24]. While much of the research into victimisation tends to explore single forms of abuse rather than multiple forms of abuse within the same study, this paper examined the risk factors of LGBTQI+ identity abuse across a wide range of abuse types. In doing so, it highlighted the high level of polyvictimisation among LGBTQI+ participants across their lifetime. Since the adverse consequences of victimisation are more pronounced when it targets one's LGBTQI+ identity and multiple forms are experienced [19,41], it is important for future research to identify the prevalence of polyvictimisation in relation to LGBTQI+ identity abuse as well as establish the ways in which it affects victims, not only in relation to their wellbeing and mental health but also broader outcomes related to substance use, education, employment, relationships, and social inclusion. Siloing research into single forms of victimisation prevents researchers and policy makers from understanding the breadth of identity abuse against the LGBTQI+ population and addressing it comprehensively though prevention and policy measures [64].

## Conclusion

Our findings demonstrate that LGBTQI+ participants not only experience high levels of LGBTQI+ identity abuse and polyvictimisation related to their gender and sexual identity over the course of their lifetime but also in the past year alone. However, the risk of each type of identity abuse varied across sexual and gender minorities highlighting that risk is constituted differently for each subgroup. The majority of transgender and gender non-conforming participants experienced gender abuse (being misnamed/misgendered). The free text comments also shone a light on the intersectional nature of abuse experiences that are rooted in ableism and racism as well as issues in relation to invisible disabilities. These results highlight the requirement for polices and measures to safeguard against identity abuse for LGBTQI+ people, such as robust hate crime/speech legislation, public education and awareness to combat prejudice, and inclusive policies and practices across education, workplace and healthcare settings. The results also point to the need for research that takes an intersectional perspective and identifies the context of the abuse in terms of the settings and perpetrators.

## Supporting information

**S1 File. Supplementary info_related manuscripts.**
(DOCX)

## Acknowledgments

We would like to acknowledge the contribution of the RAG for providing valuable guidance and the wider research team who contributed to the main study, namely, Dr. Thelma Begley, Prof. Louise Doyle, Dr. Brian Keogh, Dr. Paul Corcoran and Dr. Mark Monahan. Most importantly, we wish to thank the research participants who took the time to respond to the survey.

## Author contributions

**Conceptualization:** Carmel Downes, Jan de Vries, Matt Kennedy, Agnes Higgins.

**Data curation:** Carmel Downes, Karin O'Sullivan, Agnes Higgins.

**Formal analysis:** Carmel Downes, Karin O'Sullivan, Renee Molloy, Aviejay Paul, Agnes Higgins.

**Funding acquisition:** Agnes Higgins.

**Investigation:** Carmel Downes, Karin O'Sullivan.

**Methodology:** Carmel Downes, Jan de Vries, Matt Kennedy, Agnes Higgins.

**Writing – original draft:** Carmel Downes, Agnes Higgins.

**Writing – review & editing:** Carmel Downes, Karin O'Sullivan, Jan de Vries, Matt Kennedy, Renee Molloy, Aviejay Paul, Agnes Higgins.

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
