## [Decision Letter · Decision Letter 0]

18 Sep 2025

Dear Dr. Higgins,

Thank you for submitting your manuscript to PLOS ONE. After careful consideration, we feel that it has merit but does not fully meet PLOS ONE’s publication criteria as it currently stands. Therefore, we invite you to submit a revised version of the manuscript that addresses the points raised during the review process.

We look forward to receiving your revised manuscript.

Kind regards,

Davide Costa

Academic Editor

PLOS ONE

Journal Requirements:

2. In the online submission form, you indicated that [The participants of this study did not give written consent for their data to be shared publicly, so due to the sensitive nature of the research supporting data are not available. The data that support the findings of this study are available from the corresponding author, [AH], upon reasonable request.].

Reviewers' comments:

Reviewer's Responses to Questions

**Comments to the Author**

1. Is the manuscript technically sound, and do the data support the conclusions?

Reviewer #1: Yes

2. Has the statistical analysis been performed appropriately and rigorously?

Reviewer #1: Yes

3. Have the authors made all data underlying the findings in their manuscript fully available?

Reviewer #1: Yes

4. Is the manuscript presented in an intelligible fashion and written in standard English?

Reviewer #1: Yes

Reviewer #1: This manuscript presents a technically sound, well-designed mixed-methods study using a large national cohort to examine the prevalence and predictors of identity-based abuse among LGBTQI+ populations in Ireland. The combination of quantitative analyses (chi-square tests and multivariate logistic regression with clearly defined p-values) and qualitative analysis translates to rigorous and comprehensive research with appropriate controls. The findings robustly support the authors’ conclusions regarding high prevalence rates and subgroup differences, and the manuscript is clearly structured, logically organized, and written in standard English. Only minor improvements are suggested. Overall, the study meets methodological and reporting standards and merits publication after minimal refinements as listed below:

Line 62-66: A clear picture of what constitutes abuse, or at least how it is defined in this paper earlier on would be helpful. Table 2 (line 265) illustrates the different types of abuse and what they mean in the context of the survey, but I found myself wondering about the relationship between ‘abuse’ and related concepts like harassment, assault, violence, hate crime, and microaggression up until that point. The terms hate speech, harassment, assault, violence, and discrimination are used throughout the introduction, and it would be important to clarify whether all of those acts fall under the broader umbrella of abuse.

Line 74-83: A little bit of context regarding the referenced studies would be helpful. Even just including where the study was conducted (e.g. In these studies conducted in X and Y…) and how the study was conducted (e.g. survey, focus group, interview).

Line 86-88: What ages are included under the term young people? A general comment which I come back to later is that there is a lack of defining what ‘young’ means in the paper. Sometimes they are defined as people under 18, and sometimes they are under 35. I think clearly defining distinct terms to indicate those below 18 vs. those below 35 would be important. And like the comment above, it would be helpful to have more context about the study being mentioned here.

Line 97-98: Please provide a little more context about the referenced study.

Line 101-105: Please provide a little more context about the referenced study.

Line 116-118: Please provide a little more context about the referenced study.

Line 122-136: The rationale for including the variables are very clear, except for urbanity. Why was this included? How has urbanity affected likelihood of experiencing abuse in prior studies if there are any?

Line 162-166: What are the print and digital media (e.g. are they newsprints, magazines), and how does digital media differ from social media? Were they advertisements? Which social media platforms were used? What does interviews with regional radio stations mean? Were the researchers interviewed by radio hosts to discuss and promote the study? More detail and clarity would be nice.

Line 171-203: In lines 143-145 it is mentioned that an advisory steering committee comprised of members of the sexual and gender minority community were involved in the survey design. I think this is an important point to expand on in the methodology. Involvement of the community being studied follows principles of community-based participatory research (CBPR) which increases the validity and reliability of the study. CRPR principles are also a cornerstone of minority research, and I would add a reference to support that.

Line 199: Closing bracket without opening bracket.

Line 201-202: In line 485 it is mentioned that there is no reliable estimate of LGBTQI+ population in Ireland. I think this is important to mention here to describe why the sample representativeness is unknown. I was left wondering why it was unknown until I got to line 485.

Line 225-234: Is there a specific qualitative technique or approach that was used and could be referenced here? Thematic analysis framework developed by Braun & Clarke come to mind. Mentioning a concrete qualitative model/approach would strengthen the qualitative portion of this study.

Line 225-234: What is the role of AH in the qualitative analysis specifically? What does it mean to have a second researcher? Did they go back and review the data/codes after the discussions? Or were they just included in the discussions? Did anything change after the discussions took place and AH got involved? I have no idea what AH’s role was from the current description, and so I can’t really agree that it enhances trustworthiness of the process. I also wonder if validity and reliability are better words to use than trustworthiness.

Line 237-241: Even though the specific data is available in Table 1, I find it helpful to include percentages in the sentences directly (e.g. Nearly a third (30.2%) identified as transgender and…). Percentages are included in the latter half of the paragraph, and I would encourage the same in the first half of this paragraph.

Line 252: Figure 1 is missing.

Line 287: Related to an earlier comment, I think there needs to be a clear distinction between young people between ages 14-18 and young people aged under 35. Earlier the sample is called ‘young’ because they were under 35. Now the same word is used to describe people who 14-18. Could terms like Underaged Youth be used for people 14-18, and Young Adult for people who are up to 35 years old? Aside from clarity, I think someone who is 14 and another person who is 35 are at vastly different stages of life and would experience have different forms and degrees of abuse.

Line 293: What does the colour green in Table 3 signify? There is no description of that.

Line 300-301: Could percentages be included after the number of participants in the sentence for quick context?

Line 303: Does the term women in this case include just cis women and not trans women, as trans and gender non-confirming folk are grouped separately? If so I would use cis women consistently because using the term women at the exclusion of trans women doesn’t seem right.

Line 370-373: The higher prevalence of abuse in this study compared to EU and prior report in Ireland is quite staggering, is there any possible explanations for why that may be?

Line 390: I think a comma or semi-colon could be used here.

Line 400-404: Is there a way to separate this sentence into two? This was the first instance where I had to reread a sentence couple of times to understand it fully.

Line 402: The word ‘also’ is repeated.

Line 406: Is there a particular reason why the acronym TGNC wasn’t used here suddenly?

Line 406-407: I think TGNC and cis women should also be compared for a fuller picture/comparison between the three groups, especially since in the next sentence cisgender people are mentioned broadly.

Line 409-411: Could the word ‘that’ be replaced with ‘the’?

Line 441-443: When it’s mention that most were men, does it mean cis men compared to trans men? And is the term gay used here to include those who identify as lesbians as well? The wording in this sentence is very confusing as to who is included and who is not.

Line 444: Extra closing bracket.

Line 453: Extra closing bracket.

Line 468-472: I think the sentence can end after ‘ethnicity’ in line 470, and a new sentence can start from ‘however’. Also, I think it would be nice to mention other studies from Ireland or EU which have found that ethnicity leads to more or less exposure to abuse for context.

Line 470: Extra comma after the word data?

Line 475-490: I think the advisory steering committee should be mentioned as a strength; community participation under principles of community-based participatory research (CBPR) is known to increase validity/reliability of research.

Line 476: Extra ‘not’.

Line 478-480: What is 'more covert' abuse? What does it include and look like? Again, I think it would be important to outline what abuse means and includes/excludes in this paper earlier on.

Line 493-496: Confused as to the difference between the terms prejudiced individuals and stigmatized minority – does this sentence mean that victims of abuse are connected to others of the same identity to receive support? Could this point be clarified?

Line 497: Why is it identity harassment and not identity abuse here?

Line 498: Seems like a word is missing after ‘counter’.

Line 508-514: Is there a reason why intersectionality theory and minority stress theory weren’t mentioned/discussed earlier in the paper? I think it is very important to discuss them early as guiding theories and lenses for this paper. I find it odd that they are only introduced/mentioned briefly at the very end.

Line 536-539: Could examples of the policies and measures be provided? It doesn’t have to be in-depth – maybe just list them (e.g. education and awareness, workplace sensitivity training, inclusive language policy, etc.).

Line 542-546: Why is the contribution of the advisory steering committee not acknowledged?

**Do you want your identity to be public for this peer review?** For information about this choice, including consent withdrawal, please see our Privacy Policy

Reviewer #1: No

---

## [Author Response · Author response to Decision Letter 1]

3 Oct 2025

We have uploaded our response to the reviewer's comments in a word document.

---

## [Editor Report · Decision Letter 1]

15 Oct 2025

Identity abuse among sexual and gender minority communities: the Being LGBTQI+ in Ireland study

PONE-D-25-39928R1

Dear Dr. Higgins,

We’re pleased to inform you that your manuscript has been judged scientifically suitable for publication and will be formally accepted for publication once it meets all outstanding technical requirements.

Kind regards,

Davide Costa

Academic Editor

PLOS ONE
---

## [Editor Report · Acceptance letter]

PONE-D-25-39928R1

PLOS ONE

Dear Dr. Higgins,

I'm pleased to inform you that your manuscript has been deemed suitable for publication in PLOS ONE. Congratulations! Your manuscript is now being handed over to our production team.

Kind regards,

on behalf of

Dr. Davide Costa

Academic Editor

PLOS ONE